# Sparse convolutional coding for neuronal assembly detection

**Sven Peter**[1,*]     **Elke Kirschbaum**[1,*]
{sven.peter,elke.kirschbaum}@iwr.uni-heidelberg.de

**Martin Both**[2]                                    **Lee A. Campbell**[3]
mboth@physiologie.uni-heidelberg.de          lee.campbell@nih.gov

**Brandon K. Harvey**[3]                         **Conor Heins**[3,4,†]
bharvey@mail.nih.gov                       conor.heins@ds.mpg.de

**Daniel Durstewitz**[5]                       **Ferran Diego Andilla**[6,‡]
daniel.durstewitz@zi-mannheim.de      ferran.diegoandilla@de.bosch.com

**Fred A. Hamprecht**[1]
fred.hamprecht@iwr.uni-heidelberg.de

[1]Interdisciplinary Center for Scientific Computing (IWR), Heidelberg, Germany
[2]Institute of Physiology and Pathophysiology, Heidelberg, Germany
[3]National Institute on Drug Abuse, Baltimore, USA
[4]Max Planck Institute for Dynamics and Self-Organization, Göttingen, Germany
[5]Dept. Theoretical Neuroscience, Central Institute of Mental Health, Mannheim, Germany
[6]Robert Bosch GmbH, Hildesheim, Germany

## Abstract

Cell assemblies, originally proposed by Donald Hebb (1949), are subsets of neurons firing in a temporally coordinated way that gives rise to repeated motifs supposed to underly neural representations and information processing. Although Hebb's original proposal dates back many decades, the detection of assemblies and their role in coding is still an open and current research topic, partly because simultaneous recordings from large populations of neurons became feasible only relatively recently. Most current and easy-to-apply computational techniques focus on the identification of strictly synchronously spiking neurons. In this paper we propose a new algorithm, based on sparse convolutional coding, for detecting recurrent motifs of arbitrary structure up to a given length. Testing of our algorithm on synthetically generated datasets shows that it outperforms established methods and accurately identifies the temporal structure of embedded assemblies, even when these contain overlapping neurons or when strong background noise is present. Moreover, exploratory analysis of experimental datasets from hippocampal slices and cortical neuron cultures have provided promising results.

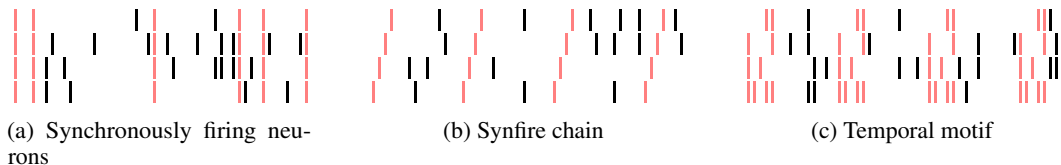

(a) Synchronously firing neurons        (b) Synfire chain        (c) Temporal motif

Figure 1: Temporal motifs in neuronal spike trains. All three illustrations show the activity of four different neurons over time. The spikes highlighted in red are part of a repeating motif. In (a) the motif is defined by the synchronous activity of all neurons, while the synfire chain in (b) exhibits sequential spiking patterns. (c) shows a more complex motif with non-sequential temporal structure. (Figure adapted from [23].)

# 1   Introduction

The concept of a *cell assembly* (or *cortical motif* or *neuronal ensemble*) was originally introduced by Donald Hebb [1] and denotes subsets of neurons that by firing coherently represent mental objects and form the building blocks of cortical information processing. Numerous experimental studies within the past 30 years have attempted to address the neural assembly hypothesis from various angles in different brain areas and species, but the concept remains debated, and recent massively parallel single-unit recording techniques have opened up new opportunities for studying the role of spatio-temporal coordination in the nervous system [2–12].

A number of methods have been proposed to identify motifs in neuronal spike train data, but most of them are only designed for strictly synchronously firing neurons (see figure 1a), i.e. with zero phase-lag [13–17], or strictly sequential patterns as in synfire chains [18–21] (see figure 1b). However, some experimental studies have suggested that cortical spiking activity may harbor motifs with more complex structure [5, 22] (see figure 1c). Only quite recently statistical algorithms were introduced that can efficiently deal with arbitrary lag constellations among the units participating in an assembly [23], but the identification and validation of motifs with complex temporal structure remains an area of current research interest.

In this paper we present a novel approach to identify motifs with any of the temporal structures shown in figure 1 in a completely unsupervised manner. Based on the idea of convolutive Non-Negative Matrix Factorization (NMF) [24, 25] our algorithm reconstructs the neuronal spike matrix as a convolution of motifs and their activation time points. In contrast to convolutive NMF, we introduce an $\ell_0$ and $\ell_1$ prior on the motif activation and appearance, respectively, instead of a single $\ell_1$ penalty. This $\ell_0$ regularization enforces more sparsity in the temporal domain; thus performing better in extracting motifs from neuronal spike data by reducing false positive activations. Adding the $\ell_0$ and $\ell_1$ penalty terms requires a novel optimization scheme. This replaces the multiplicative update rules by a combination of discrete and continuous optimizations, which are matching pursuit and LASSO regression. Additionally we added a sorting and non-parametric threshold estimation method to distinguish between real and spurious results of the optimization problem. We benchmark our approach on synthetic data against Principal Component Analysis (PCA) and Independent Component Analysis (ICA) as the most widely used methods for motif detection, and against convolutive NMF as the method most closely related to the proposed approach. Our algorithm outperforms the other methods especially when identifying long motifs with complex temporal structure. We close with results of our approach on two real-world datasets from hippocampal slices and cortical neuron cultures.

# 2   Related work

PCA is one of the simplest methods that has been used for a long time to track cell motifs [26]. Its biggest limitations are that different assembly patterns can easily be merged into a single 'large' component, and that neurons shared between motifs are assigned lower weights than they should have. Moreover, recovering individual neurons which belong to a single assembly is not reliably possible [27, 17], and the detected assemblies are not very robust to noise and rate fluctuations [23].

ICA with its assumption of non-Gaussian and statistically independent subcomponents [28] is able to recover individual neuron-assembly membership, and neurons belonging to multiple motifs are

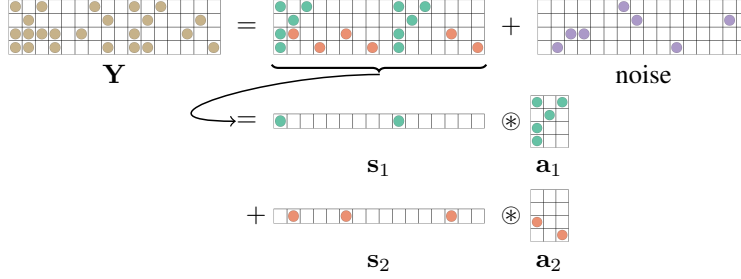

Figure 2: Sketch of convolutional coding. In this example the raw data matrix $\mathbf{Y}$ is described by a matrix which is an additive mixture of two motifs $\mathbf{a}_1$ (cyan) and $\mathbf{a}_2$ (salmon) convolved with their activities $\mathbf{s}_1$ and $\mathbf{s}_2$, respectively, plus background noise.

also correctly identified [17]. ICA provides a better estimate for synchronous motifs than PCA [17], but motifs with more complicated temporal structure are not (directly) accommodated within this framework. An overview of PCA and ICA for identifying motifs is provided in [17].

More sophisticated statistical approaches have been developed, like unitary event analysis [13, 14], for detecting coincident, joint spike events across multiple cells. More advanced methods and statistical tests were also designed for detecting higher-order correlations among neurons [15, 16], as well as synfire chains [20]. However, none of these techniques is designed to detect more complex, non-synchronous, non-sequential temporal structure. Only quite recently more elaborate statistical schemes for capturing assemblies with arbitrary temporal structure, and also for dealing with issues like non-stationarity and different time scales, were advanced [23]. The latter method works by recursively merging sets of units into larger groups based on their joint spike count probabilities evaluated across multiple different time lags. The method proposed in this paper, in contrast, approaches the detection of complex assemblies in a very different manner, attempting to detect complex patterns as a whole.

NMF techniques have been widely applied to recover spike trains from calcium fluorescence recordings [29–35]. Building on these schemes, NMF has been used to decompose a binned spike matrix into multiple levels of synchronous patterns which describe a hierarchical structuring of the motifs [36]. But these previous applications of NMF considered only neurons firing strictly synchronously. In audio processing, convolutive NMF has been successfully used to detect motifs with temporal structure [24, 25, 37]. However, as we will show later, the constraints used in audio processing are too weak to extract motifs from neuronal spike data. For this reason we propose a novel optimization approach using sparsity constraints adapted to neuronal spike data.

## 3 Sparse convolutional coding

We formulate the identification of motifs with any of the temporal structures displayed in figure 1 as a convolutional matrix decomposition into motifs and their activity in time, based on the idea behind convolutive NMF [24, 25], and combined with the sparsity constraints used in [34]. We use a novel optimization approach and minimize the reconstruction error while taking into account the sparsity constraints for both motifs and their activation time points.

Let $\mathbf{Y} \in \mathbb{R}_+^{n \times m}$ be a matrix whose $n$ rows represent individual neurons with their spiking activity binned to $m$ columns. We assume that this raw signal is an additive mixture of $l$ motifs $\mathbf{a}_i \in \mathbb{R}_+^{n \times \tau}$ with temporal length $\tau$, convolved with a sparse activity signal $\mathbf{s}_i \in \mathbb{R}_+^{1 \times m}$ plus noise (see figure 2).

We address the unsupervised problem of simultaneously estimating both the coefficients making up the motifs $\mathbf{a}_i$ and their activities $\mathbf{s}_i$. To this end, we propose to solve the optimization problem

$$\min_{\mathbf{a},\mathbf{s}} \left\| \mathbf{Y} - \sum_{i=1}^{l} \mathbf{s}_i \circledast \mathbf{a}_i \right\|_F^2 + \alpha \sum_{i=1}^{l} \|\mathbf{s}_i\|_0 + \beta \sum_{i=1}^{l} \|\mathbf{a}_i\|_1 \tag{1}$$

with $\alpha$ and $\beta$ controlling the regularization strength of the $\ell_0$ norm of the activations and the $\ell_1$ norm of the motifs, respectively. The convolution operator $\circledast$ is defined by

$$\mathbf{s}_i \circledast \mathbf{a}_i = \sum_{j=1}^{\tau} \mathbf{a}_{i,j} \cdot S(j-1)\mathbf{s}_i \tag{2}$$

with $\mathbf{a}_{i,j}$ being the $j$th column of $\mathbf{a}_i$. The column shift operator $S(j)$ moves a matrix $j$ places to the right while keeping the same size and filling missing values appropriately with zeros [24]. The product on the right-hand side is an outer product.

In [25] the activity of the learned motifs is regularized only with a $\ell_1$ prior which is too weak to recover motifs in neuronal spike trains. Instead we choose the $\ell_0$ prior for $\mathbf{s}_i$ since it has been successfully used to learn spike trains of neurons [34]. For the motifs themselves a $\ell_1$ prior is used to enforce only few non-zero coefficients while still allowing exact optimization [38].

## 3.1 Optimization

This problem is non-convex in general but can be approached by initializing the activities $\mathbf{s}_i$ randomly and using a block coordinate descent strategy [39, Section 2.7] to alternatingly optimize for the two variables.

When keeping the activations $\mathbf{s}_i$ fixed, the motif coefficients $\mathbf{a}_i$ are learned using LASSO regression with non-negativity constraints [40] by transforming the convolution with $\mathbf{s}_i$ to a linear set of equations by using modified Toeplitz matrices $\tilde{\mathbf{s}}_i \in \mathbb{R}^{mn \times n\tau}$ which are then stacked column-wise [41, 38]:

$$\min_{\mathbf{a}} \left\| \underbrace{\text{vec}(\mathbf{Y})}_{b \in \mathbb{R}^{mn}} - \underbrace{[\tilde{\mathbf{s}}_1 \quad ... \quad \tilde{\mathbf{s}}_l]}_{A \in \mathbb{R}^{mn \times ln\tau}} \underbrace{\begin{bmatrix} \text{vec}(\mathbf{a}_1) \\ ... \\ \text{vec}(\mathbf{a}_l) \end{bmatrix}}_{x \in \mathbb{R}^{ln\tau}} \right\|_2^2 + \beta \sum_{i=1}^{l} \|\mathbf{a}_i\|_1 \tag{3}$$

The matrices $\tilde{\mathbf{s}}_i$ are constructed from the $\mathbf{s}_i$ with $\tilde{\mathbf{s}}_{i,j,k} = \tilde{\mathbf{s}}_{i,j+1,k+1} = \mathbf{s}_{i,j-k}$ for $j \geq k$ and $\tilde{\mathbf{s}}_{i,j,k} = 0$ for $j < k$ and $\tilde{\mathbf{s}}_{i,j,k} = 0$ for $j > p \cdot m$ and $k < p \cdot \tau$ for $p = 1, \ldots, n$ (where $i$ denotes the $i$th matrix with element indices $j$ and $k$).

When keeping the currently found motifs $\mathbf{a}_i$ fixed, their activation in time is learned using a convolutional matching pursuit algorithm [42–44] to approximate the $\ell_0$ norm. The greedy algorithm iteratively includes an assembly appearance that most reduces the reconstruction error. All details of the algorithm are outlined in the supplementary material for this paper.

## 3.2 Motif sorting and non-parametric threshold estimation

The list of identified motifs is expected to also contain false positives which do not appear repeatedly in the data. The main non-biological reason for this is that our algorithm only finds local minima of the optimization problem given by equation (1). Experiments on various synthetic datasets showed that motifs present at the global optimum should always have the same appearance, independent of the random initialization of the activities. The false positives which are only present in particular local minima, however, look differently every time the initialization is changed. We therefore propose to run our algorithm multiple times on the same data with the same parameter settings but with different random initializations, and use the following sorting and non-parametric threshold estimation algorithm in order to distinguish between true (reproducible) and spurious motifs. The following is only a brief description. More details are given in the supplementary material.

In the first step, the motifs found in each run are sorted using pairwise matching. The sorting is necessary because the order of the motifs after learning is arbitrary and it has to be assured that the motifs with the smallest difference between different runs are compared. Sorting the sets of motifs from all runs at the same time is an NP hard multidimensional assignment problem [45]. Therefore, a greedy algorithm is used instead. It starts by sorting the two sets of motifs with the lowest assignment cost. Thereafter, the remaining sets of motifs are sorted one by one according to the order of motifs given by the already sorted sets.

Inspired by permutation tests, we estimate a threshold $T$ by creating a shuffled spike matrix to determine which motifs are only spurious. In the shuffled matrix all temporal correlations between

and within neurons have been destroyed. Hence, there are no real motifs in the shuffled matrix and the motifs learned from this matrix will likely be different with each new initialization. We take the minimal difference of any two motifs from different runs of the algorithm on the shuffled matrix as the threshold. We assume that motifs that show a difference between different runs larger than this threshold are spurious and discard them.

## 3.3 Parameter selection

The sparse convolutional coding algorithm has only three parameters that have to be specified by the user: the maximal number of assemblies, the maximal temporal length of a motif, and the penalty $\beta$ on the $\ell_1$ norm of the motifs. The number of assemblies to be learned can be set to a generous upper limit since the sorting method assures that only the true motifs remain while all false positives are deleted. The temporal length of a motif can also be set to a generous upper bound. To find an adequate $\ell_1$ penalty for the assemblies, different values need to be tested, and it should be set to a value where neither the motifs are completely empty nor all neurons are active over the whole possible length of the motifs. In the tested cases the appearance of the found motifs did not change drastically while varying the $\ell_1$ penalty within one order of magnitude, so fine-tuning it is not necessary. Instead of specifying the penalty $\alpha$ on the $\ell_0$ norm of the activations directly, we chose to stop the matching pursuit algorithm when adding an additional assembly appearance increases the reconstruction error or when the difference of reconstruction errors from two consecutive steps falls below a small threshold.

All code for the proposed method is available at: `https://github.com/sccfnad/Sparse-convolutional-coding-for-neuronal-assembly-detection`

# 4 Results

## 4.1 Synthetic data

Since ground truth datasets are not available, we have simulated different synthetic datasets to establish the accuracy of the proposed method, and compare it to existing work.

For PCA and ICA based methods the number of motifs is estimated using the Marchenko-Pastur eigenvalue distribution [17]. The sparsity parameter in the sparse convolutive NMF (scNMF) that resulted in the best performance was chosen empirically [25].

An illustrative example dataset with twenty neurons, one hundred spurious spikes per neuron and three temporal motifs can be seen in figure 3. Consecutive activation times between motifs were modeled as Poisson renewal processes with a mean inter-event-distance of twenty frames. When running our method from two different random initial states to identify a total of five motifs, all three original motifs were among those extracted from the data (figure 3c and 3d; the motifs have been sorted manually to match up with the ground truth; all parameters for the analysis can be found in table 1). While the two spurious motifs change depending on the random initialization, the three true motifs consistently show up in the search results. Neither PCA, ICA nor scNMF were able to extract the true motifs (see figures 3e, 3f and 3g).

For further analysis, various datasets consisting of fifty neurons observed over one thousand time frames were created. Details on the generation of these datasets can be found in the supplementary material. For each of the different motif lenghts $\tau = 1$, 7 and 21 frames, twenty different datasets were created, with different noise levels and numbers of neurons shared between assemblies.

To compare the performance of different methods, we use the functional association between neurons as an indicator [27, 46, 12]. For this a neuron association matrix (NAM) is calculated from the learned motifs. The NAM contains for each pair of neurons a 1 if the two neurons belong to the same assembly and a 0 otherwise. The tested methods, however, do not make binary statements about whether a neuron belongs to an assembly, but provide only the information to what degree the neuron was associated with an assembly. We apply multiple thresholds to binarize the output of the tested methods and compute true positive rate and false positive rate between the ground truth NAM and the binarized NAM, leading to the ROC curves shown in figure 4. We chose this method since it works without limitations for synchronous motifs and also allows for comparisons for the more complex cases.

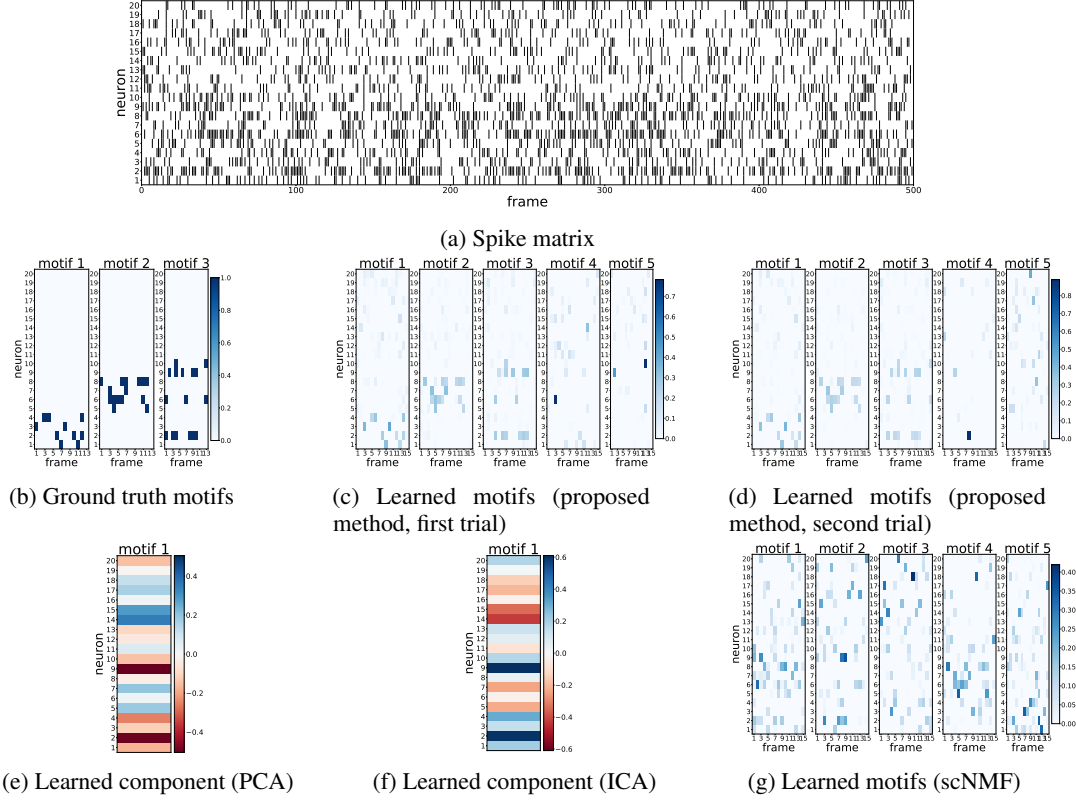

(a) Spike matrix

(b) Ground truth motifs

(c) Learned motifs (proposed method, first trial)

(d) Learned motifs (proposed method, second trial)

(e) Learned component (PCA)

(f) Learned component (ICA)

(g) Learned motifs (scNMF)

Figure 3: Results on a synthetic dataset. (a) shows a synthetic spike matrix. (b) shows the three motifs present in the data. By running our algorithm with two different random initial states the motifs seen in (c) and (d) are learned. (e), (f) and (g) show the results from PCA, ICA and scNMF, respectively.

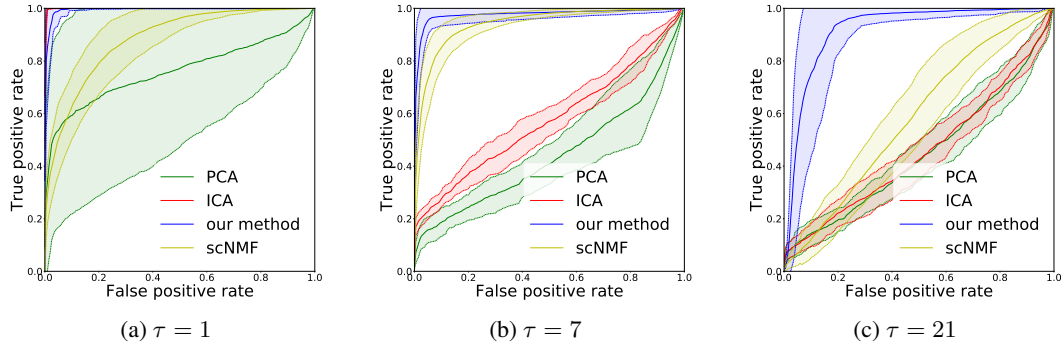

(a) $\tau = 1$

(b) $\tau = 7$

(c) $\tau = 21$

Figure 4: ROC curves of different methods on synthetic data for different temporal motif lengths. We show the mean ROC curve and its standard deviation averaged over all trials on different synthetic datasets. All methods were run ten times on each dataset with different random initializations.

In the synchronous case (i.e. $\tau = 1$, figure 4a) our proposed method performs as good as the best competitor. As expected PCA performance shows a huge variance since some of the datasets contain neurons shared between multiple motifs and since extracting actual neuron-assembly assignments is not always possible [27, 17]. When temporal structure is introduced we are still able to identify associations between neurons with very high accuracy. For short temporal motifs ($\tau = 7$, figure 4b) scNMF is able to identify associations, but only our method was able to accurately recover most associations in long motifs ($\tau = 21$, figure 4c).

Table 1: Experimental parameters. We show the used maximal number of assemblies, maximal motif length in frames, $\ell_1$ penalty value $\beta$, and number of runs of the algorithm with different initializations for the performed experiments on synthetic and real datasets. We also display the estimated threshold $T$ used for distinguishing between real and spurious motifs.

| Experiment | #motifs | motif length in frames | $\beta$ | #runs | $T$ |
|---|---|---|---|---|---|
| synthetic example data | 5 | 15 | $5 \cdot 10^{-4}$ | 2 | – |
| hippocampal CA1 region | 5 | 10 | $10^{-6}$ | 5 | $5.7 \cdot 10^{-6}$ |
| cortical neuron culture | 5 | 10 | $10^{-6}$ | 5 | $6.5 \cdot 10^{-4}$ |

## 4.2 Real data

**In vitro hippocampal CA1 region data.** We analyzed spike trains of 91 cells from the hippocampal CA1 region recorded at high temporal and multiple single cell resolution using $CA^{2+}$ imaging. The acute mouse hippocampal slices were recorded in a so-called interface chamber [47].

On this dataset, our algorithm identified three motifs as real motifs. They are shown in figure 5a. The activity of each assembly has been calculated at every frame and is shown in figure 5b. In order to qualitatively show that the proposed method appropriately eliminates false positives from the list of found motifs also on real data, we plotted in figure 6 for each motif the difference to the best matching motif from every other run. We did this for the motifs identified in the original spike matrix (figure 6a), as well as for the motifs identified in the shuffled spike matrix (figure 6b). The motifs found in the shuffled matrix show much higher variability between runs than those found in the original matrix. For motifs 1 and 3 from the original matrix the difference between runs is in average about two to three times higher than for the other motifs, but still smaller than the average difference between runs for all of the motifs from the shuffled data. Nevertheless, these motifs are deleted as false positives, since the threshold for discarding a motif is set to the minimum difference of motifs from different runs on the shuffled matrix. This shows that the final set of motifs is unlikely to contain spurious motifs anymore.

The spontaneous hippocampal network activity is expected to appear under the applied recording conditions as sharp wave-ripple (SPW-R) complexes that support memory consolidation [48–50, 47]. Motif 5 in figure 5a shows the typical behavior of principal neurons firing single or two consecutive spikes at a low firing rate ($\ll 1\,\mathrm{Hz}$) during SPW-R in vitro [47]. This might be interpreted as the re-activation of a formerly established neuronal assembly.

**In vitro cortical neuron culture data.** Primary cortical neurons were prepared from E15 embryos of Sprague Dawley rats as described in [51] and approved by the NIH Animal Care and Usage Committee. Cells were transduced with an adeno-associated virus expressing the genetically-encoded calcium indicator GCaMP6f on DIV 7 (Addgene #51085). Wide-field epifluorescent videos of spontaneous calcium activity from individual wells ($6 \times 10^4$ cells/well) were recorded on DIV 14 or 18 at an acquisition rate of 31.2 frames per second. The data for the shown example contains 400 identified neurons imaged for 10 minutes on DIV 14.

Our algorithm identified two motifs in the used dataset, shown in figure 5c. Their activity is plotted in figure 5d. For each column of the two motifs, figure 7 shows the percentage of active neurons at every time frame. The motifs were thresholded such that only neurons with a motif coefficient above 50% of the maximum coefficient of the motif were counted. We show those columns of the motifs which contained more than one neuron after thresholding. The fact that figure 7 shows only few motif activations that include all of the cells that are a part of the motif has less to do with the actual algorithm, but more with how the nervous system works: Only rarely all cells of an assembly will spike [23], due to both the intrinsic stochasticity, like probabilistic synaptic release [52] and the fact that synaptic connectivity and thus assembly membership will be graded and strongly fluctuates across time due to short-term synaptic plasticity [53]. Nevertheless, the plot shows that often several columns are active in parallel and there are some time points where a high percentage of the neurons in all columns is active together. This shows that the found motifs really contain temporal structure and are repeated multiple times in the data.

All parameters for the analysis of the shown experiments can be found in table 1.

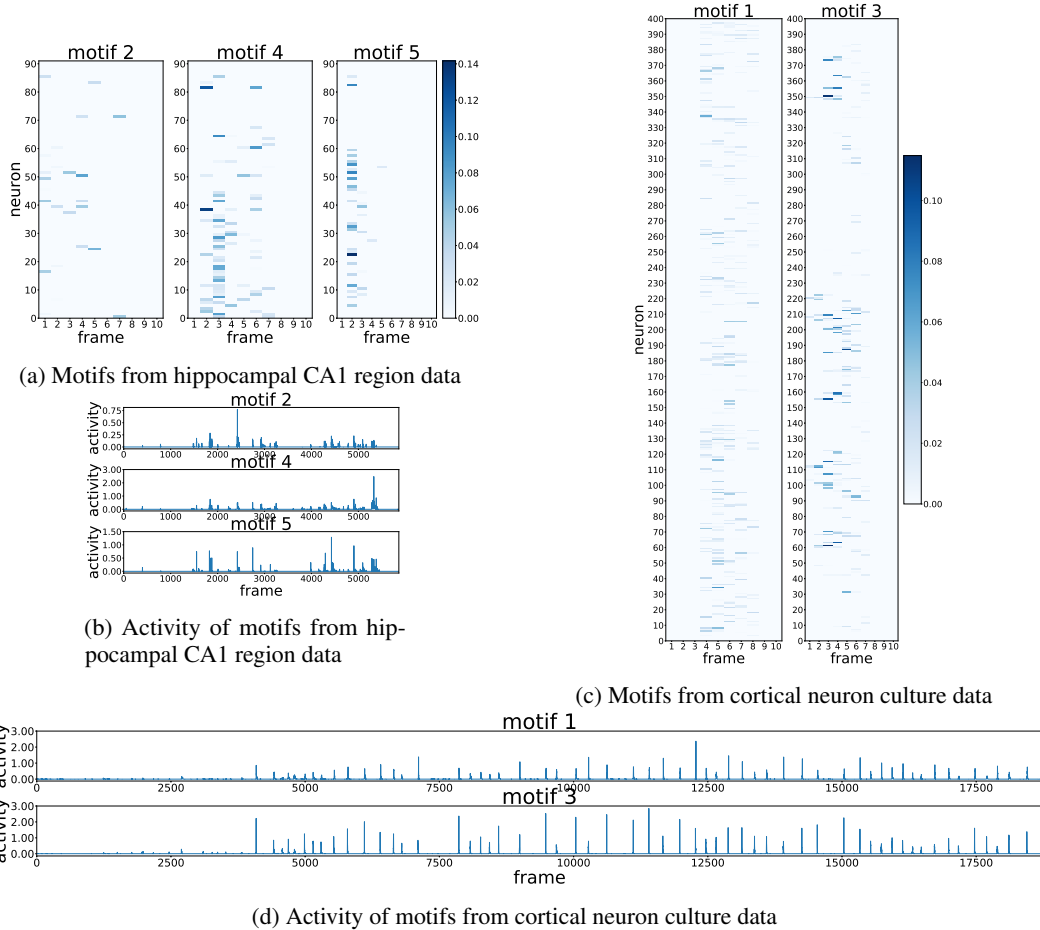

(a) Motifs from hippocampal CA1 region data

(b) Activity of motifs from hippocampal CA1 region data

(c) Motifs from cortical neuron culture data

(d) Activity of motifs from cortical neuron culture data

Figure 5: Results from real data. We show the results of our algorithm for two different real datasets. The datasets vary in temporal length as well as number of observed cells. For each dataset we show the motifs that our algorithm identified as real motifs and their activity over time.

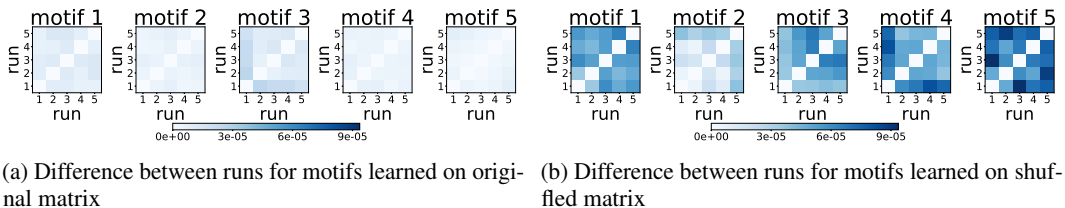

(a) Difference between runs for motifs learned on original matrix

(b) Difference between runs for motifs learned on shuffled matrix

Figure 6: Differences between the five runs for all five learned motifs from hippocampal CA1 region data. The plots show for each motif the difference to the best matching motif from every other run. We did this for the motifs identified in the original hippocampal CA1 region data (a), as well as for the motifs identified in the shuffled spike matrix (b). The motifs found in the shuffled matrix show much higher variability between runs than those found in the original matrix.

## 5 Discussion

We have presented a new approach for the identification of motifs that is not limited to synchronous activity. Our method leverages sparsity constraints on the activity and the motifs themselves to allow a simple and elegant formulation that is able to learn motifs with temporal structure. Our algorithm extends convolutional coding methods with a novel optimization approach to allow modeling of interactions between neurons. The proposed algorithm is designed to identify motifs in data with temporal stationarity. Non-stationarities in the data, which are expected to appear especially in

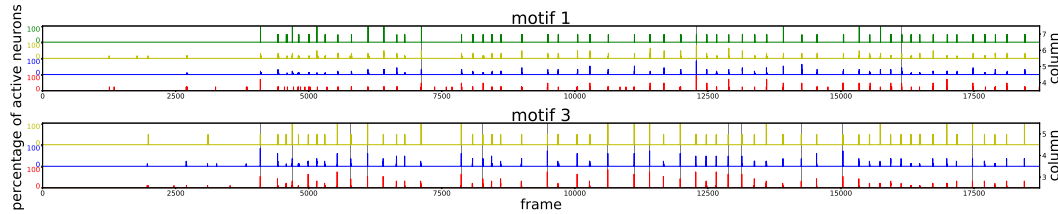

Figure 7: Percentage of active neurons per column over time, for all motifs identified in the cortical neuron culture dataset. For each column of the two motifs displayed in figure 5c, we show the percentage of active neurons at every time frame. Vertical grey bars indicate points in time at which all significantly populated columns of a motif fire with at least 30% of their neurons. Their reoccurence shows that the motifs really contain temporal structure and are repeated multiple times in the dataset.

recordings from in vivo, are not yet taken into account. In cases where non-stationarities are expected to be strong, the method for stationarity-segmentation introduced in [54] could be used before applying our algorithm to the data. Although our algorithm has some limitations in terms of non-stationarities, results on simulated datasets show that the proposed method outperforms others especially when identifying long motifs. Additionally, the algorithm shows stable performance on real datasets. Moreover, the results found on the cortical neuron culture dataset show that our method is able to detect assemblies within large sets of recorded neurons.

### Acknowledgments

SP and EK thank Eleonora Russo for sharing her knowledge on generating synthetic data and Fynn Bachmann for his support. LAC, BKH and CH thank Lowella Fortuno for technical assistance with cortical cultures and acknowledge the support by the Intramural Research Program of the NIH, NIDA. DD acknowledges partial financial support by DFG Du 354/8-1. SP, EK, MB, DD, FD and FAH gratefully acknowledge partial financial support by DFG SFB 1134.

## Footnotes

†Majority of this work was done while co-author was at [3].

‡Majority of this work was done while co-author was at [1].

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
