[Supplementary Material]

# Supplementary material for the paper: *Sparse convolutional coding for neuronal assembly detection*

**Sven Peter**[1,*]     **Elke Kirschbaum**[1,*]
{sven.peter,elke.kirschbaum}@iwr.uni-heidelberg.de

**Martin Both**[2]                          **Lee A. Campbell**[3]
mboth@physiologie.uni-heidelberg.de          lee.campbell@nih.gov

**Brandon K. Harvey**[3]                     **Conor Heins**[3,4,†]
bharvey@mail.nih.gov                         conor.heins@ds.mpg.de

**Daniel Durstewitz**[5]                     **Ferran Diego Andilla**[6,‡]
daniel.durstewitz@zi-mannheim.de             ferran.diegoandilla@de.bosch.com

**Fred A. Hamprecht**[1]
fred.hamprecht@iwr.uni-heidelberg.de

[1]Interdisciplinary Center for Scientific Computing (IWR), Heidelberg, Germany
[2]Institute of Physiology and Pathophysiology, Heidelberg, Germany
[3]National Institute on Drug Abuse, Baltimore, USA
[4]Max Planck Institute for Dynamics and Self-Organization, Göttingen, Germany
[5]Dept. Theoretical Neuroscience, Central Institute of Mental Health, Mannheim, Germany
[6]Robert Bosch GmbH, Hildesheim, Germany

## 1   Optimization

We want to solve the following optimization problem

$$\min_{\mathbf{a},\mathbf{s}} \left\| \mathbf{Y} - \sum_{i=1}^{l} \mathbf{s}_i \circledast \mathbf{a}_i \right\|_F^2 + \alpha \sum_{i=1}^{l} \|\mathbf{s}_i\|_0 + \beta \sum_{i=1}^{l} \|\mathbf{a}_i\|_1 \tag{1}$$

where $\circledast$ denotes the convolution operator which is defined by

$$\mathbf{s}_i \circledast \mathbf{a}_i = \sum_{j=1}^{\tau} \mathbf{a}_{i,j} \cdot S(j-1)\mathbf{s}_i \tag{2}$$

with $\mathbf{a}_{i,j}$ being the $j$th column of $\mathbf{a}_i$ and $\mathbf{Y} \in \mathbb{R}_+^{n \times m}$. The column shift operator $S(j)$ moves a matrix $j$ places to the right while keeping the same size and filling missing values appropriately with zeros

---

[*]Both authors contributed equally.
[†]Majority of this work was done while co-author was at [3].
[‡]Majority of this work was done while co-author was at [1].

[1], for example:

$$\mathbf{A} = (a \quad b \quad c \quad d)$$
$$S(0)\mathbf{A} = (a \quad b \quad c \quad d)$$
$$S(1)\mathbf{A} = (0 \quad a \quad b \quad c)$$
$$S(2)\mathbf{A} = (0 \quad 0 \quad a \quad b) \quad .$$

The optimization problem in equation (1) is non-convex in general but can be solved approximately by initializing the $\mathbf{s}_i$ randomly and using a block coordinate descent strategy [2, Section 2.7] to alternatingly optimize for the two variables.

## 1.1 Update the motifs $\mathbf{a}_i$

When keeping the assembly activations $\mathbf{s}_i$ fixed, the motif coefficients $\mathbf{a}_i$ can be learned using LASSO regression with non-negativity constraints [3] by transforming the convolution with $\mathbf{s}_i$ to a linear set of equations using modified Toeplitz matrices $\tilde{\mathbf{s}}_i \in \mathbb{R}^{mn \times n\tau}$ which are then stacked next to each other [4, 5]:

$$\min_{\mathbf{a}} \left\| \mathrm{vec}(\mathbf{Y}) - \sum_{i=1}^{l} \tilde{\mathbf{s}}_i \cdot \mathrm{vec}(\mathbf{a}_i) \right\|_2^2 + \beta \sum_{i=1}^{l} \|\mathbf{a}_i\|_1$$

$$= \min_{\mathbf{a}} \left\| \underbrace{\mathrm{vec}(\mathbf{Y})}_{b \in \mathbb{R}^{mn}} - \underbrace{[\tilde{\mathbf{s}}_1 \quad \dots \quad \tilde{\mathbf{s}}_l]}_{A \in \mathbb{R}^{mn \times ln\tau}} \underbrace{\begin{bmatrix} \mathrm{vec}(\mathbf{a}_1) \\ \dots \\ \mathrm{vec}(\mathbf{a}_l) \end{bmatrix}}_{x \in \mathbb{R}^{ln\tau}} \right\|_2^2 + \beta \sum_{i=1}^{l} \|\mathbf{a}_i\|_1 \quad (3)$$

The matrices $\tilde{\mathbf{s}}_i$ are constructed from the $\mathbf{s}_i$ with $\tilde{\mathbf{s}}_{i,j,k} = \tilde{\mathbf{s}}_{i,j+1,k+1} = \mathbf{s}_{i,j-k}$ for $j \geq k$ and $\tilde{\mathbf{s}}_{i,j,k} = 0$ for $j < k$ and $\tilde{\mathbf{s}}_{i,j,k} = 0$ for $j > p \cdot m$ and $k < p \cdot \tau$ for $p = 1, \dots, n$ (where $i$ denotes the $i$th matrix with element indices $j$ and $k$, $\tau$ is the number of columns of $\mathbf{a}_i$, and $n$ and $m$ are the number of rows and columns of the data matrix, respectively). That means the matrix $\tilde{\mathbf{s}}_i$ consists of $n$ blocks $\mathfrak{s}_i$ with

$$\mathfrak{s}_i = \left. \begin{pmatrix} s_{i,0} & 0 & \dots & 0 \\ s_{i,1} & s_{i,0} & \ddots & \vdots \\ \vdots & s_{i,1} & \ddots & 0 \\ \vdots & \vdots & & s_{i,0} \\ \vdots & \vdots & & \vdots \\ s_{i,m} & s_{i,m-1} & \dots & s_{i,m-\tau} \end{pmatrix} \right\} m \qquad (4)$$

$$\underbrace{\qquad\qquad\qquad}_{\tau}$$

and

$$\tilde{\mathbf{s}}_i = \left. \begin{pmatrix} \boxed{\mathfrak{s}_i} & & & 0 \\ & \boxed{\mathfrak{s}_i} & \overset{n \text{ times}}{\searysep} & \\ & & \ddots & \\ 0 & & & \boxed{\mathfrak{s}_i} \end{pmatrix} \right\} mn \qquad . \qquad (5)$$

$$\underbrace{\qquad\qquad\qquad}_{n\tau}$$

Special care has to be taken to avoid missing parts of the motif due to the originally identified positions. Consider the ground truth motif shown in figure 1a. After a single iteration the learned

(a) Ground truth motif
(b) Possible wrong motifs after a single iteration
(c) Equivalent padded motifs

Figure 1: Ground truth motif and learned state after a single iteration. This figure shows four neurons forming a motif over three frames as shown in (a). After a single iteration parts of the motif can be missing as shown in (b), which is solved in (c) by using a larger assembly length and centering the motifs after each iteration.

---

**Data:** binned spike matrix $\mathbf{Y} \in \mathbb{R}_+^{n \times m}$, $\forall i \in [1, ..., l]$: assembly $\mathbf{a}_i \in \mathbb{R}_+^{n \times \tau}$
**Result:** $\forall i \in [1, ..., l]$: updated assembly activity $\mathbf{s}_i \in \mathbb{R}_+^{1 \times m}$
initialize all elements of $\mathbf{s}_i$ to zeros $\forall i$;
**while** *not converged* **do**
    calculate current residual $\mathbf{R} = \mathbf{Y} - \sum_i^l \mathbf{s}_i \circledast \mathbf{a}_i$;
    initialize inner product result $\mathbf{P} \in \mathbb{R}_+^{l \times m}$ to zeros;
    **foreach** *frame* $f \in [1, ..., m]$ **do**
        **foreach** *assembly index* $i \in [1, ..., l]$ **do**
            **foreach** *offset* $j \in [0, ..., \tau - 1]$ **do**
                $u \leftarrow$ column $j$ of assembly $\mathbf{a}_i$ ;
                $v \leftarrow$ column $f + j$ of residual $\mathbf{R}$ ;
                $\mathbf{P}_{if} \leftarrow \mathbf{P}_{if} + u^T v$ ;
            **end**
        **end**
    **end**
    $i^*, f^* \leftarrow \arg \max \mathbf{P}$ ;
    increase $f^*$-th element of $\mathbf{s}_{i^*}$ by $\max \mathbf{P}$ ;
**end**

**Algorithm 1:** Convolutional matching pursuit algorithm to learn the assembly activities while keeping the currently found motifs fixed. After initializing $\mathbf{s}_i$ to zeros, assembly appearances are added wherever they reduce the reconstruction error. The algorithm stops when adding an additional assembly appearance increases this error or when the error is converged.

motif could be any of the two wrong possibilities seen in figure 1b. While the learned motif does indeed occur in the data, it is not complete and can never be completed since there is no more space on the left or right to identify the missing associations. To overcome this problem the vectors $\mathbf{a}_i$ have to be chosen larger than required and centered after each iteration when possible: When there are enough empty columns on either side the whole motif is shifted before the new assembly activities $\mathbf{s}_i$ are learned. This does not increase the reconstruction error since the activities will also just be shifted by the same amount. When new coefficients $\mathbf{a}_i$ are learned in the next iteration there now is enough space to also capture the previously missed associations (see figure 1c which can be completed in the next iteration). Another possibility is to move the center of mass of the motif. This method showed also good results, especially on datasets with high noise levels.

## 1.2 Update the activations $\mathbf{s}_i$

When keeping the currently found motifs $\mathbf{a}_i$ fixed, their activation in time is learned using a convolutional matching pursuit algorithm [6–8] to approximate the $\ell_0$ norm. The greedy algorithm iteratively chooses to include the assembly appearance that most reduces the reconstruction error given the current residual. The pursuit stops when adding an additional assembly appearance increases this error or when the difference of this error to the error in the previous step falls below a small threshold. Algorithm 1 explains this approach in detail.

---

**Data:** binned spike matrix $\mathbf{Y} \in \mathbb{R}_+^{n \times m}$, upper bound $l$ on the number of assemblies and $\tau$ on their length

**Result:** $\forall i \in [1, ..., l]$: assembly $\mathbf{a}_i \in \mathbb{R}_+^{n \times \tau}$, assembly activity $\mathbf{s}_i \in \mathbb{R}_+^{1 \times m}$

$\forall i \in [1, ..., l]$: randomly initialize all elements of $\mathbf{s}_i$ to 0 or 1;

**while** *not converged* **do**

    $\mathbf{a} \leftarrow$ LASSO solution of equation 3 depending on $\mathbf{Y}$ and $\mathbf{s}$;

    **foreach** *assembly* $\mathbf{a}_i$ **do**

        **if** *number of empty columns on both sides is not balanced* **then**

            $\mathbf{a}_i \leftarrow \overset{\xi \rightarrow}{\mathbf{a}_i}$ or $\mathbf{a}_i \leftarrow \overset{\leftarrow \xi}{\mathbf{a}_i}$ to balance empty columns

        **end**

    **end**

    $\mathbf{s} \leftarrow$ matching pursuit approximation (algorithm 1) using $\mathbf{Y}$ and $\mathbf{a}$;

    **foreach** *assembly activity vector* $\mathbf{s}_i$ **do**

        **if** $\|\mathbf{s}_i\| \leq$ *small threshold* $\epsilon$ **then**

            randomly initialize all elements of $\mathbf{s}_i$ to 0 or 1;

        **end**

    **end**

**end**

---

**Algorithm 2:** Sparse convolutional coding algorithm. After initializing the activities of the assemblies to random noise, a block coordinate descent strategy is used to alternatingly learn the motifs with LASSO regression and their activities with a convolutional matching pursuit algorithm. After each iteration the motifs are centered if possible and activities below a small threshold are newly initialized to random noise.

Sometimes a given assembly $\mathbf{a}_i$ completely disappears during the matching pursuit because all its components are better explained by another assembly $\mathbf{a}_j$. In this case the assembly activity vector $\mathbf{s}_i$ is randomly reinitialized to allow another different assembly to appear in the next iteration.

Based on these considerations, algorithm 2 is proposed to learn assemblies and their activity vectors simultaneously given a spike matrix.

## 2 Motif sorting and non-parametric threshold estimation

As our algorithm only finds local minima of the optimization problem in equation (1), the list of motifs is expected to contain false positives that are not actually present in the spike data. Unlike real motifs, which should always have the same appearance, the random initialization of the activities will cause the false positives to look differently every time it is changed. We therefore propose to run our algorithm multiple times on the same data with the same parameter settings in order to distinguish between persistent and spurious motifs. The following method is used to find out which motifs reappear in different runs.

### 2.1 Motif sorting – K-partite matching

The motifs found in each run have to be sorted because the order of the motifs after learning is arbitrary and it has to be assured that the motifs with the greatest similarity are compared between the different runs. Therefore, the motifs within each set have to be ordered in such a way that the difference between all runs is minimized for all motifs.

Let $K$ be the number of times the algorithm was run with the same parameters but different initializations (the number of trials), $l$ the upper bound on the number of motifs and let $\mathbf{a}_i^k$ be the $i$th motif found during the $k$th run. This will result in $K$ sets $\{\mathbf{a}_i^k | i = 1, ..., l\}$ for $k = 1, \ldots, K$ each of which contains the motifs of a single trial. Given that the order of the motifs in each trial (and thus their indices $i$) is arbitrarily chosen and therefore independent, we want to find permutations $\pi_k : \{1, \ldots, l\} \rightarrow \{1, \ldots, l\}$ of these indices for each $k$ such that $\mathbf{a}_{\pi_k(i)}^k$, the $i$th motif in the $k$th trial, is most similar to the $i$th motif in all other permuted trials. To this end we propose to solve the

following optimization problem:

$$\min_{\pi_2,\dots,\pi_K} \sum_{i=1}^{l} \sum_{k=1}^{K} \sum_{p=k+1}^{K} d\left(\mathbf{a}^p_{\pi_p(i)}, \mathbf{a}^k_{\pi_k(i)}\right) \tag{6}$$

where $d(\mathbf{x}, \mathbf{y})$ denotes the difference between two motifs defined as

$$d(\mathbf{x}, \mathbf{y}) = \min_{j} \frac{\|S(j)\mathbf{x} - \mathbf{y}\|_2^2}{\|\mathbf{x}\|_0 \cdot \|\mathbf{y}\|_0} \qquad . \tag{7}$$

We allow $j = -\tau, \dots, \tau$ to also include shifts to the left. Dividing by the product of the $\ell_0$ norms of the two motifs assures that a reasonable comparison of the difference between two motifs with only few spikes and the difference between two motifs with many spikes is still possible.

The sorting of the motifs is a $K$-dimensional assignment problem also known as K-partite matching which can be solved exactly in polynomial time for $K = 2$ using the Hungarian algorithm [9, 10] but is NP hard in general [11]. An approximate solution is found by using a greedy algorithm that starts by first finding the two trials that have the lowest assignment cost after permutation. Afterwards, the remaining sets of motifs are sorted one after the other according to the order of motifs given by the already sorted sets.

Once all sets are sorted, for each motif $i$ the distance of the results from the $K$ trials to their medoid $\mathbf{M}_i$ is computed. The medoid is the $i$th motif from trial $m_i$ that has the minimal difference to the $i$th motif in all other trials and is defined by

$$m_i = \arg\min_{p} \sum_{k=1}^{K} d\left(\mathbf{a}^p_{\pi_p(i)}, \mathbf{a}^k_{\pi_k(i)}\right) \qquad , \tag{8}$$

$$\mathbf{M}_i = \mathbf{a}^{m_i}_{\pi_{m_i}(i)} \qquad . \tag{9}$$

If the distance $d(\mathbf{M}_i, \mathbf{a}^k_{\pi_k(i)})$ of a motif from its medoid is larger than a threshold $T$, the motif $\mathbf{a}^k_{\pi_k(i)}$ is erased from the list of representatives of motif $i$. Motifs where only one representative is left (the medoid itself) are assumed to be spurious results of the approximate solution of equation (1) and discarded completely.

In order to also get rid of spurious spikes and find the final set of motifs $\mathbf{a}^*_i$, we use for every motif coefficient $\mathbf{a}^*_{i,\nu t}$ the minimal coefficient value of all selected representatives of the motif:

$$\mathbf{a}^*_{i,\nu t} = \min\left\{\mathbf{a}^k_{\pi_k(i),\nu t} \,|\, k \in \{1,\dots,K\} \wedge d\left(\mathbf{M}_i, \mathbf{a}^k_{\pi_k(i)}\right) < T\right\} \tag{10}$$

where $\nu$ and $t$ are the neuron (row) and time (column) index of the motif.

## 2.2 Non-parametric threshold estimation

To determine the threshold $T$ above which motifs are discarded as false positives, we use a shuffled spike matrix. This matrix is created by shuffling each row of the original spike matrix independently to preserve the number of spikes per neuron but destroy any temporal correlations between and within neurons [12]. On a matrix with no temporal correlations there should be no matching motifs in-between runs. Therefore, the shuffled matrix is also analyzed using the sparse convolutional coding algorithm with $K$ random initializations and the same parameters as used for the analysis of the original matrix. The motifs $\tilde{\mathbf{a}}^k_i$ found in the shuffled matrix are also sorted as described in section 2.1.

The threshold $T$ is set to be the minimal distance between any motif $\tilde{\mathbf{a}}^k_{\pi_k(i)}$ found in the shuffled matrix and the corresponding medoid $\mathbf{M}_i$

$$T = \min_{\substack{i \in \{1,\dots,l\} \\ k \in \{1,\dots,K\}, k \neq m_i}} d\left(\tilde{\mathbf{a}}^k_{\pi_k(i)}, \mathbf{M}_i\right) \qquad . \tag{11}$$

## 2.3 Exemplary result on synthetic data

We show the results for the above described algorithm on an exemplary set of synthetic data.

(a) Spike matrix          (b) Ground truth motifs

Figure 2: Exemplary synthetic dataset with three motifs. (a) shows the spike matrix for a set of twenty neurons observed over one thousand time frames. In addition to the three motifs shown in (b) this dataset also contains fifty spurious spikes.

This dataset consists of twenty neurons observed over one thousand time frames (spike matrix see figure 2a). A subset of the neurons is randomly assigned to belong to a single motif, others to multiple motifs and the rest are not part of any assembly and fire completely on their own. The assembly activity itself is modeled as a Poisson process with a randomly chosen mean [13] and a refractory period of at least the length of the assembly itself. Additionally spurious spikes of single neurons are added to simulate neurons firing out of sync. In the shown example three motifs appear in the data (see figure 2b).

To distinguish real from spurious motifs, the sparse convolutional coding algorithm is applied four times to the dataset as well as to the shuffled spike matrix. In every run we were looking for five motifs with a length of ten frames. The $\ell_1$ penalty $\beta$ was set to $10^{-4}$. The set of motifs found in each of the four runs is shown in figure 3a to 3d.

In a first step the motifs in each set are sorted such that the $i$th motif in the first trial is most similar to the $i$th motif in all other trials as described above. The resorted sets are shown in figure 4. Figure 5 shows the matched motifs.

In the same fashion the motifs found in the shuffled matrix (shown in figure 6) are sorted and the matched motifs are shown in figure 7. As all temporal correlation between neurons has been destroyed, there are no repeating motifs in the shuffled matrix. For this reason we expect to find different motifs whenever we change the initialization. Hence, we can use the distance between motifs from different runs on the shuffled matrix as a measure to distinguish whether motifs found in different runs on the original matrix are similar. For this reason we define the threshold $T$ as the minimal distance of a motif found in runs of the algorithm with different initializations on the shuffled matrix to its medoid. For each motif (except for the medoid itsself) the distance to its medoid is computed and the minimum of these values is taken to be $T$. In this example we find $T = 2.81 \cdot 10^{-5}$.

Once the sorting is done and the threshold is determined, for each of the five motifs found on the original matrix the medoid over the four runs is calculated and motifs with a difference to the medoid larger than the threshold $T = 2.81 \cdot 10^{-5}$ are excluded from the list of real motifs. The remaining representatives for each of the five originally learned motifs are shown in figure 8. Motifs for which only the medoid itself is left are considered to be spurious motifs and are therefore deleted. Finally, persistent motifs are found by taking for all motif coefficients the minimum value over the remaining representatives of the motif. The resulting motifs for this example are shown in figure 9.

## 3    Generation of synthetic datasets

Various datasets consisting of fifty neurons observed over one thousand time frames were created for the comparison of our approach to some well-established methods. A subset of the neurons is randomly assigned to belong to a single motif, others to multiple motifs and the rest are not part of any assembly and fire completely on their own. The assembly activity itself is modeled as a Poisson

Table 1: Experimental parameters. We show the used maximal number of assemblies, maximal motif length in frames, $\ell_1$ penalty value $\beta$, and number of runs of the algorithm with different initializations for the performed experiments on different hippocampal CA1 region and cortical neuron culture datasets. We also display the estimated threshold $T$ used for distinguishing between real and spurious motifs.

| Experiment | #motifs | motif length in frames | $\beta$ | #runs | $T$ |
|---|---|---|---|---|---|
| hippocampal CA1 region | | | | | |
| dataset 5 | 5 | 10 | $10^{-6}$ | 5 | $1.49 \cdot 10^{-5}$ |
| dataset 15 | 5 | 10 | $10^{-6}$ | 5 | $9.24 \cdot 10^{-6}$ |
| cortical neuron culture | | | | | |
| dataset L97_P1 | 5 | 10 | $10^{-6}$ | 5 | $1.23 \cdot 10^{-2}$ |
| dataset L97_P3 | 5 | 10 | $10^{-7}$ | 5 | $2.26 \cdot 10^{-6}$ |

Table 2: Runtimes. CPU runtime for the analysis of different parts of the cortical neuron culture dataset with parameter settings as shown in the paper.

| analyzed part of the dataset | #neurons | #time frames | runtime in minutes |
|---|---|---|---|
| complete dataset | 400 | 18733 | 425 |
| first half of the time frames | 400 | 9366 | 192 |
| second half of the time frames | 400 | 9366 | 206 |
| first quarter of time frames | 400 | 4683 | 72 |
| last quarter of time frames | 400 | 4683 | 117 |
| half of the neurons, all time frames | 200 | 18733 | 154 |
| one quarter of the neurons, all time frames | 100 | 18733 | 64 |

process with a randomly chosen mean [13] and a refractory period of at least the length of the motif itself. Additionally spurious spikes of single neurons are added to simulate neurons firing out of sync. The percentile of neurons belonging to multiple motifs and the fraction of spurious spikes have been varied to create different test cases. For each of the three temporal motif lengths $\tau = 1, 7$ and $21$ frames twenty different datasets were created.

# 4 Results on real data

We analyzed several datasets which were recorded similarly to those described in the paper. Figure 10 shows two examples of results of our method on additional hippocampal CA1 region datasets. Figure 11 shows results from two additional datasets from cortical neuron culture data. The parameters used for the analysis of these datasets can be found in table 1. The identified motifs from these datasets show similar temporal structures as the motifs shown in the paper.

# 5 Runtime

General statements about runtime are difficult, since it depends not only on the size of the data, but also on the choice of parameters (maximum motif length, maximum number of motifs, ensemble penalty, number of initialisations) as well as on the sparsity of the data and implementational details. To give a rough intuition for the dependence on neuron number and data length, we did the analysis for different slices of the cortical neuron culture dataset shown in the paper (400 neurons, 18733 time frames) with the same parameter settings. We show always the CPU time for one run of the sparse convolutional coding algorithm on an Intel(R) Xeon(R) CPU E5-2650 v3 @ 2.30GHz machine. This dataset is very sparse at the beginning (approximately the first quarter of the time frames) and shows increased firing activity in the rest of the dataset. In order to reduce computation costs, the implementation of the conversion into Toeplitz matrices follows equations (4) and (5), resulting in the LASSO regression being the bottleneck w.r.t. runtime.

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

(a) Motifs found in run 1

(b) Motifs found in run 2

(c) Motifs found in run 3

(d) Motifs found in run 4

Figure 3: Sets of found motifs for each of the four initializations on original spike matrix. Shown are the results from the sparse convolutional coding algorithm on the synthetic dataset for four runs with different random initializations. In each run we learned five motifs with a length of ten frames and $\ell_1$ penalty $\beta = 10^{-4}$.

(a) Sorted motifs in run 1

(b) Sorted motifs in run 2

(c) Sorted motifs in run 3

(d) Sorted motifs in run 4

Figure 4: Sorted sets for each of the four runs on original spike matrix. The motifs were sorted such that the difference of the $i$th motif in the first run to the $i$th motif in all other runs is minimized.

(a) Motif 1

(b) Motif 2

(c) Motif 3

(d) Motif 4

(e) Motif 5

Figure 5: Matched motifs from original spike matrix. We show for each of the five motifs the motif from each of the four runs that matches best.

(a) Motifs found in run 1

(b) Motifs found in run 2

(c) Motifs found in run 3

(d) Motifs found in run 4

Figure 6: Sets of found motifs for each of the four initializations on shuffled matrix. Shown are the results from the sparse convolutional coding algorithm on the shuffled dataset for four runs with different random initializations. In each run we learned five motifs with a length of ten frames and $\ell_1$ penalty $\beta = 10^{-4}$.

(a) Motif 1

(b) Motif 2

(c) Motif 3

(d) Motif 4

(e) Motif 5

Figure 7: Matched motifs from shuffled matrix. We show for each of the five motifs the motif from each of the four runs that matches best.

(a) Remaining representatives of motif 1

(b) Remaining representatives of motif 2

(c) Remaining representatives of motif 3

(d) Remaining representatives of motif 4

(e) Remaining representatives of motif 5

Figure 8: Remaining representatives of the motifs from original spike matrix. For each of the five motifs the medoid over the four runs is calculated and motifs with a difference to the medoid bigger than $T = 2.81 \cdot 10^{-5}$ are deleted from the list of representatives for this motif. Shown are the remaining representatives for each of the five motifs.

Figure 9: Final set of motifs. The final set of motifs is achieved by taking for all motif coefficients the minimum value over the remaining representatives of the motif. Motifs where only one representative is left, are deleted. In the shown case only three motifs have remained, which are in almost perfect agreement with the ground truth motifs.

(a) Motifs from hippocampal CA1 region dataset 5

(b) Activity of motifs from hippocampal CA1 region dataset 5

(c) Motifs from hippocampal CA1 region dataset 15

(d) Activity of motifs from hippocampal CA1 region dataset 15

Figure 10: Additional results from in vitro hippocampal CA1 region data. We show four examples of motifs (left) found in different hippocampal CA1 region datasets and the motifs activity in time (right).

(a) Motifs from cortical neuron culture dataset L97_P1

(b) Motifs from cortical neuron culture dataset L97_P3

(c) Activity of motifs from cortical neuron culture dataset L97_P1

(d) Activity of motifs from cortical neuron culture dataset L97_P3

Figure 11: Additional results from in vitro cortical neuron culture data. We show two examples of motifs (top) found in different cortical neuron culture datasets and the motifs activity in time (bottom).