[Reviews · NeurIPS 2017]

Reviewer 1



The paper proposes a new method for the identification of spiking motifs in neuronal population data. The authors’ approach is to use a sparse convolutional coding model that is learned using an alternation between LASSO and matching pursuit. The authors address the problem of false positives and compare their algorithm to 3 competing methods in both simulation experiments and real neural data. It appears that the basic model is not original, but that the regularization has been improved, leading to a method that performs well beyond the state of the art. I believe that the method could be of some use to the neuroscience community as it may be powerful enough to determine if population codes fire in coordinated assemblies and if so, what is the nature of the assemblies? This is a step towards addressing these long-standing questions in the neuroscience community. Overall, the paper is clearly written, the method is reasonably conceived and described, and the simulation studies were sufficiently conceived for a paper of this length. Some of the methods for evaluating performance were unclear. I am not totally certain how to assess Figure 4, for example since lines 169-175 where the “functional association” metric is described is too impoverished to determine what was actually done and what is being compared across detection thresholds. I also am of the opinion that the authors are over-interpreting their data analysis results when they claim that the motif appearance is related to re-activation of former assemblies (do they mean replay?). Also, it appears that few of the motif activations in Figure 7 include all of the cells that are a part of the motif. This is concerning as it is not a part of the model description and makes it appear as though there is a poor model fit to the data.

Reviewer 2



This submission introduces an algorithm to extract temporal spike trains in neuronal assemblies, based on a sparse convolutive non-negative matrix factorization (scNMF). This algorithm is evaluated on synthetic datasets and compares favorably with PCA, ICA and scNMF. The difference with scNMF is not obvious and, despite the supplementary material, the update rules for the motifs and the activities are not obvious. The text mentions a difference in the regularization scheme, it should be emphasized. The contribution could be reinforced, the interests of the algorithm could be better explained by formally describing the difference with state of the art. Another remark is about the name of the algorithm and the article title: why changing the name "sparse convolutive" NMF to "convolutional"? Without specific reason, it may be more adequate to keep the terminology "sparse convolutive" to keep the consistency with the existing literature.

Reviewer 3



* Summary The paper "Sparse convolutional coding for neuronal assembly detection" proposes a new approach to detecting stationary occurring spike motifs across neurons. The proposed algorithm uses a non-negative matrix factorization of the data matrix into "occurrences" and "motifs". The authors test their algorithm on synthetic and real data from cell cultures, compare their algorithm against PCA and ICA, and give strategies to identify spurious motifs and determine the best parameters of the algorithm. * Comments The algorithm is a nice application of non-negative matrix factorization (NMF). The paper is generally well written, and the algorithm and the mathematics look sound. The algorithm seems to work well on synthetic data and produce stable results on real data. The only additional point I would have been interested in, is run time, in particular the maximal time length that can be run in reasonable time and how the algorithm scales with the number of neurons. For instance, the optimization problem (3) seems very costly when converting the activity signals into Toeplitz matrices. However, the matrices are very sparse, so the algorithm might still scale well. I few words on that would be good. * Minor Comments - All plots should be re-generated (esp. figures 3 and 4) because all fonts are basically unreadable.